# Nanoplatforms Potentiated Ablation-Immune Synergistic Therapy through Improving Local Control and Suppressing Recurrent Metastasis

**DOI:** 10.3390/pharmaceutics15051456

**Published:** 2023-05-10

**Authors:** Zixuan Wei, Xiaoya Yu, Mao Huang, Liewei Wen, Cuixia Lu

**Affiliations:** 1Medical College, Guangxi University, Nanning 530004, China; qiangkunxuan@163.com (Z.W.); yuxiaoya0108@163.com (X.Y.); 2Guangdong Provincial Key Laboratory of Tumor Interventional Diagnosis and Treatment, Zhuhai People’s Hospital (Zhuhai Hospital Affiliated with Jinan University), Jinan University, Zhuhai 519000, China; huangm43@mail2.sysu.edu.cn

**Keywords:** tumor ablation, immunotherapy, nanoparticles, synergistic anti-tumor effect

## Abstract

Minimally invasive ablation has been widely applied for treatment of various solid tumors, including hepatocellular carcinoma, renal cell carcinoma, breast carcinomas, etc. In addition to removing the primary tumor lesion, ablative techniques are also capable of improving the anti-tumor immune response by inducing immunogenic tumor cell death and modulating the tumor immune microenvironment, which may be of great benefit to inhibit the recurrent metastasis of residual tumor. However, the short-acting activated anti-tumor immunity of post-ablation will rapidly reverse into an immunosuppressive state, and the recurrent metastasis owing to incomplete ablation is closely associated with a dismal prognosis for the patients. In recent years, numerous nanoplatforms have been developed to improve the local ablative effect through enhancing the targeting delivery and combining it with chemotherapy. Particularly, amplifying the anti-tumor immune stimulus signal, modulating the immunosuppressive microenvironment, and improving the anti-tumor immune response with the versatile nanoplatforms have heralded great application prospects for improving the local control and preventing tumor recurrence and distant metastasis. This review discusses recent advances in nanoplatform-potentiated ablation-immune synergistic tumor therapy, focusing on common ablation techniques including radiofrequency, microwave, laser, and high-intensity focused ultrasound ablation, cryoablation, and magnetic hyperthermia ablation, etc. We discuss the advantages and challenges of the corresponding therapies and propose possible directions for future research, which is expected to provide references for improving the traditional ablation efficacy.

## 1. Introduction

Tumor ablation is an imaging-guided, minimally invasive surgical procedure that is commonly used to treat solid tumors such as liver, kidney, lung, and brain tumors. During such a procedure, a needle-like probe guided by ultrasound, CT, or MRI is inserted into a solid tumor. Chemotherapy, alcohol, microwaves, radio waves, laser, or a freezing agent are delivered through this probe to kill the tumor cells [1,2,2,3,4]. The imaging-guided procedure was initially performed for diagnosis by needle biopsy but exploited for direct therapeutic intervention shortly after. In 1983, Sugiura and colleagues described the direct injection of absolute ethanol into small hepatocellular carcinoma (HCC) under ultrasound guidance [5], and this work was soon followed by the direct injection of chemotherapy into solid tumors of the liver, pancreas, pelvis, and lung [6,7,8,9,10,11]. In 1993, Rossi and co-workers first used radiofrequency ablation (RFA) to treat small HCC [12]. Microwave ablation (MWA) was first performed by Seki et al. in 1994 to treat HCC of less than 2 cm [13]. Cryoprobes were used to treat lesions as early as the 1960s [14,15,16], and the technology development over recent decades has expanded their clinical applications in oncotherapy, including treatment of prostate cancer [17,18,19,20,21,22]. Compared with chemotherapy, radiotherapy, and conventional surgery, tumor ablation has the advantages of being less invasive, more effective and safer, more cost-efficient, and having a larger indication range [23,24,25,26,27]. Currently, tumor ablation is included in the guidelines for the treatment of a variety of solid tumors (Figure 1). RFA, MWA, cryoablation, and anhydrous alcohol have been recommended as local treatment for liver cancer [28,29,30]. Ablation has also become a well-established local therapy for metastatic or recurrent tumors to achieve NED (tumor-free state) or relieve symptoms and cancerous bone pain [31,32,33,34]. Although ablation therapy is an effective method for the local destruction of tumor masses, it remains a palliative measure in the treatment of advanced and disseminated tumors [35]. However, it is becoming increasingly evident that this limitation can be rectified by the use of intra-tumor immunostimulants to enhance the potential anti-tumor immune response induced by ablation therapy [36,37,38]. Unlike surgical resection of the tumor, ablation interventions result in in situ coagulative tumor necrosis with the release of tumor antigens and danger-signaling molecules, which in turn may potentially activate or enhance the body’s anti-tumor immune response.

However, tumors have developed mechanisms to create an immunoreactive tumor microenvironment (TME); thus, the anti-tumor immune response stimulated by ablation alone is generally modest. There is increasing evidence according to which the localized delivery of immunomodulators can boost ablation-induced anti-tumor immunity [39]. Accordingly, tumor ablation in combination with localized immunotherapy has increasingly been used to treat various benign and malignant tumors in preclinical and clinical research [29,40]. Nanoplatforms are macromolecular materials composed of nanoparticles such as nanotubes, nanosheets, and mesoporous materials. They can be used as drug carriers for treating cancers [41,42,43,44]. Multifunctional nanoplatforms can deliver multiple drugs with optimized drug encapsulation and therapeutic efficiency. In the context of cancer treatment, nanoparticle-based drug delivery systems have shown great potential in improving treatment efficacy and reducing toxicity. Furthermore, the ability of nanoparticles to enhance drug accumulation in tumor tissues and the immune system makes them an attractive platform for enhancing ablation-immune synergistic therapy. Other than serving as drug carriers, nanoplatforms themselves have radiotherapy potentiation capability, photothermal and photodynamic effects, magnetic hyperthermia effects, and immunomodulatory properties [45,46]. Nanoparticles can modulate the immunosuppressive TME to induce and enhance immune responses. They can also be used as adjuvants for cancer vaccines to boost vaccine-induced anti-tumor immune responses. Recent research demonstrates the efficacy of immunomodulatory biomaterials in preclinical models of cancer and autoimmune diseases and discusses the potential of these materials for use in combination with other immunotherapeutic approaches [47]. Furthermore, nanoplatforms can be used in imaging techniques to identify and locate tumors, guide ablation procedures, and monitor drug response (Table 1). These characteristics make nanoplatforms an ideal material for integrating chemotherapy, immunotherapy, and ablation therapy to achieve synergistic anti-cancer outcomes.

## 2. Classification and Characteristics of Tumor Ablation Therapy

In recent years, cancer treatment has seen tremendous progress with the development of novel therapeutic strategies such as ablation and immunotherapy. Ablation involves the physical destruction of cancer cells through methods such as heat, cold, or radiation, while immunotherapy aims to stimulate the patient’s immune system to fight cancer. However, these approaches are not always effective on their own, and can have limitations such as toxicity and resistance (Table 2). To overcome these challenges, researchers have been exploring the use of nanoplatforms to deliver both ablation and immunotherapy agents simultaneously. This collaborative strategy has shown promising results in preclinical studies and holds great potential for the development of more effective cancer treatments.

### 2.1. Radiofrequency Ablation

During an RFA procedure, radiofrequency waves are sent through the probe into the surrounding tumor tissues, creating an alternating electric current that heats up and kills the tumor cells by coagulative necrosis. Each RFA treatment is about 30 min long, and the necrotic tissues are usually absorbed in situ. Compared with conventional tumor treatment methods, RFA is safer, less painful, and more cost-efficient. Additionally, patients usually have a fast recovery and can be discharged after 1–2 days of observation. However, RFA has limited effects on large tumors, and its anti-tumor efficacy is often affected by tumor heterogeneity [61,62]. For this reason, RFA is usually combined with other therapeutic modalities for integrative treatment of cancer [63]. For example, Li and colleagues created RFA-responsive cisplatin cross-linking poly(N-isoprylacrylamide-b-methyl acrylic acid) (PNA) nanogels (Pt-PNA) for triple therapies of ablation–chemo–embolization [49] (Figure 1A,B). Cisplatin is a popular metal ion-based antitumor drug that also exhibits RF-responsive thermal effects through the coordinate bonding between Pt (II) ions and carboxyl groups. The Pt-PNA nanogels can kill tumor cells through both the cytotoxic action of cisplatin and an RFA-induced thermal effect. In addition, the nanogels can rapidly spread to the tumor peripheral arteries and block the tumor blood supply elicited by temperature-sensitive sol–gel conversion to achieve transcatheter arterial chemoembolization (TACE) (Figure 1C–E). Yang and colleagues constructed unique pH-sensitive nanoreactors by co-encapsulating lipoxidase (LOX) and heme chloride with poly (lactic-co-glycolic acid) (PLGA) (HLCaP) using a CaCO_3_-assisted double emulsion method. These nanoreactors can utilize RFA-generated tumor debris as the fuel to continuously produce cytotoxic lipid radicals, inducing immunogenic cell death (ICD) and inhibiting the growth of residual tumors following RFA [48] (Figure 1F,G). Cao and colleagues found that radio frequency can potentially serve as a safe and effective physical adjuvant to boost vaccination [64]. In murine models of intradermal vaccination, RFA treatment of a small skin area (2 × 2 cm^2^) prior to intradermal delivery of ovalbumin or recombinant hemagglutinin can enhance vaccine-induced humoral and cellular immune responses with potency comparable to common chemical adjuvants. These adjuvant effects may be attributed to increased vaccine uptake and the maturation of dendritic cells (DCs) in the skin and draining lymph nodes (DLNs) following RFA (Figure 1H,I).

### 2.2. High-Intensity Focus Ultrasound

In ultrasound ablation (UA), also known as high-intensity focused ultrasound (HIFU), many ultrasound beams, guided by MRI or ultrasound, focus on the exact tumor area that requires treatment. The highly focused ultrasound energy heats up the targeted tumor tissues and causes coagulative necrosis of the tumor cells. The resulting tumor cell debris can activate the body’s immune responses, inducing tumor ICD. The ability of HIFU to pass through overlying skin and tissues without damage has made it an attractive precision therapy for deep-seated tumors. However, the ultrasound energy decreases exponentially with the depth of the tissue, leading to insufficient intensity for certain deep-seated tumors. This problem can be remedied by introducing nanoparticles into tumor tissues to change the acoustic environment. Chen and colleagues designed and synthesized multifunctional organic–inorganic hybrid nano-vesicles of PLGA loaded with perfluorocarbon (PFC), the hydrophobic anti-tumor drug ruthenium complex (RuPOP), and superparamagnetic nano Fe_3_O_4_ [65]. The nano-vesicles were coated with an ultrathin silica shell to achieve double therapies of HIFU and HIFU-triggered explosive release of RuPOP. Hypoxia within the TME contributes to immune resistance of tumor cells [66] and limit the therapeutic efficacy of HIFU. To overcome this problem, Liang and colleagues constructed ultra-stable PFC nanodroplets (D-vPCs-O_2_) to co-deliver oxygen and doxorubicin (DOX) to the target tumor [67]. The nanodroplets had an atomic polyorganosiloxane surface and pH-sensitive tumor-targeting peptide for tumor-specific cargo delivery (Figure 2A). HIFU was used to trigger the co-release of DOX and oxygen while enhancing ultrasound imaging, thereby enabling imaging-guided drug delivery. Mild-temperature HIFU (M-HIFU) not only triggered oxygen release, but also slightly increased tumor temperature and blood flow, further ameliorating tumor hypoxia. The chemotherapy and relief of hypoxia jointly downregulated the expression of transforming growth factor-β 1 (TGF-β1), thereby reducing the epithelial–mesenchymal transition (EMT) and inhibiting tumor metastasis (Figure 2B). When D-vPCs-O_2_ was used with M-HIFU as a neoadjuvant, heat shock proteins were significantly downregulated, and tumor recurrence after high-temperature HIFU (H-HIFU) ablation was reduced. In clinical practice, HIFU as a stand-alone therapy cannot induce sufficient anti-tumor immune response to eradicate tumor metastasis or recurrence. Strategies to boost HIFU efficacy for inhibiting tumor metastasis and recurrence are under active development. Kuai and colleagues developed perfluorooctyl bromide (PFOB) nano-emulsions containing MnO_2_ nanoparticles (MBPs) that can amplify HIFU-induced tumor ICD (Figure 2C) [50]. By simultaneously depleting the glutathione (GSH) and boosting HIFU-induced tumor ICD, MBPs can modulate the tumor immune microenvironment by inducing dendritic cell (DC) maturation and promoting CD8^+^ and CD4^+^ T cell activation, thereby inhibiting tumor growth and lung metastasis (Figure 2D–F).

### 2.3. Microwave Ablation

Microwave ablation (MWA) uses heat generated from electromagnetic microwaves to induce the coagulative necrosis of tumor cells. MWA alone usually triggers a weaker immune response than RFA, possibly due to high temperature-induced tumor-associated antigen (TAA) inactivation. Nevertheless, compared with RFA, MWA has significant advantages because it can have deeper tissue penetration, better efficacy for larger tumors, and reduced sensitivity to the heat sink effect [68]. Hence, various strategies have been developed to enhance the anti-tumor effects of MWA. Zhu’s team proposes a Ca^2+^-surplus alginate hydrogel plus MWA treatment to ablate tumors on both mice and rabbits, and to elicit antitumor immunity when synergized with Mn^2+^. The metallo-alginate hydrogel acts as a microwave-susceptible and immunostimulatory biomaterial to reinforce the MWA therapy, which is promising for clinical translation [69]. Carbon dots (CDs) are carbon-based nanomaterials with fluorescence properties. Due to their excellent biocompatibility and abundant low-cost sources, CDs have found important applications in biomedicine [70]. Zhou and colleagues used mannose to synthesize CDs with a diameter of about 3 nm [52]. In in vitro experiments, these mannose-derived CDs effectively captured TAAs after MWA treatment and delivered them to DCs, promoting DC maturation and the secretion of cytokines such as IL-1β, IL-6, and TNF-α (Figure 3A–D). In mice, the intra-tumor injection of these CDs after MWA significantly inhibited liver metastasis and produced long-term immune memory to inhibit tumor recurrence. Hydrogels have excellent biological properties and are widely used as carriers for drug or cell delivery. Due to the fact that hydrogels can encapsulate TAAs or immunotherapeutic drugs for controlled release and promote the infiltration of immune cells, they can be used to enhance the efficacy of immunotherapy [71]. Due to the space confinement effect, the metal ions encapsulated in hydrogels can effectively boost MWA. Microwaves can rapidly heat up hydrogels without affecting the surrounding area; thus, the surrounding tissues are well protected during MWA. Cao and colleagues were the first to report an injectable, microwave-sensitive hydrogel with immunomodulatory properties, which boosted anti-tumor immunity elicited by combined MWA and immunotherapy [72]. The immunomodulatory hydrogel was prepared by introducing various immunostimulants and an immunoadjuvant (R837) into the alginate (ALG)-Ca^2+^ hydrogel. The loaded R837 enhanced in situ vaccination with TAAs released after MWA and promoted DC maturation, eliciting potent anti-tumor immunity and suppressing distant metastasis in vivo (Figure 3E,F). Li and colleagues showed that the rapid tumor progression following inadequate MWA is mainly mediated by the so-called “cold” tumor immune environment, which is characterized by the enrichment of immunosuppressive factors and a lack of cytotoxic T lymphocytes infiltration [53]. In the same study, Li and colleagues designed and constructed a reactive oxygen species (ROS)-sensitive in situ hydrogel-based scaffold for the co-delivery of the selective PI3Kγ inhibitor IPI549 and an anti-programmed death ligand 1 antibody (aPDL1) for post-ablation cancer immunotherapy. IPI549 can reverse the immunosuppressive niche created by inadequate MWA and promote immune checkpoint blockade (ICB)-mediated anti-tumor immune response. The IPI549-loaded hydrogel can reshape the tumor immune microenvironment (TIME) by reducing the presence of CD11b^+^ immunosuppressive cells (including myeloid-derived suppressor cells and tumor-associated macrophages (TAMs)) and enhancing the infiltration of CD8^+^ T cells (Figure 3G,H). In combination with the continuous release of aPDL1, the ROS-responsive scaffold can trigger a strong anti-cancer immune response to create a “hot” tumor immune niche, degenerating/eradicating primary tumors, inhibiting the development of distant and diffuse metastases, and eliciting immunologic memory that provides long-lasting protection. Additionally, to combat the high invasiveness of residual tumor following inadequate MWA, Shen and colleagues developed a fibrin hydrogel scaffold for the co-delivery of IPI549 and the immunostimulatory chemotherapy Oxaliplatin (OX) for post-ablation cancer treatment [73]. In mouse CT26 colorectal cancer models, the IPI549 and OX-loaded hydrogel evoked robust systemic anti-tumor immunity to inhibit tumor growth and metastasis, and induced strong long-term immunologic memory against tumor recurrence (Figure 3I).

### 2.4. Magnetic Hyperthermia Ablation

Magnetic hyperthermia ablation (MHA) uses magnetic responsive materials to transform magnetic energy into thermal energy under the action of an alternating magnetic field to kill tumor cells. The mechanisms for thermal energy generation include eddy current loss, hysteresis loss, Néel relaxation, and Brownian relaxation. The magnetic field used in MHA has no tissue penetration limit, and does not produce any thermal effect on tissues in the absence of magnetic responsive materials. Thus, MHA is a suitable therapy for deep-seated tumors, including large ones [74,75]. However, due to the difficulties in precisely targeting deep-seated tumors and achieving homogeneous heat distribution within the tumor, the clinical application of MHA is relatively rare [76,77]. PAN and colleagues synthesized monodisperse, high-performance superparamagnetic CoFe_2_O_4_@MnFe_2_O_4_ nanoparticles with a diameter of about 11.9 nm for MHA of primary tumors [54]. Using these nanoparticles, MHA can trigger the release of abundant TAAs to promote the maturation and activation of DCs and cytotoxic T cells, delivering effective immunotherapy for distant metastatic tumors in tumor-bearing mice (Figure 4).

### 2.5. Cryoablation

In cryoablation, a thin, hollow needle called a cryoprobe is placed directly into the target tumor. A freezing agent such as liquid nitrogen is pumped into the cryoprobe to freeze and kill the tumor cells by necrosis or apoptosis. Compared with RFA and MWA, cryoablation is less painful and has the benefit of a faster postoperative recovery [78]. Cryoablation is a relatively new option in oncotherapy and is most suitable for the treatment of elderly patients with tumors near the skin. Cryoablation also produces TAAs in situ that can activate the host’s anti-tumor immunity and thus, it can serve as an adjuvant to enhance immunotherapy [79,80]. However, cryoablation has a few limitations. In clinical practice, cryoablation often fails to freeze and kill the entire target tumor due to the presence of a temperature gradient between the cryoprobe and the tumor tissue. In addition, surrounding healthy tissues may suffer cryogenic injury caused by off-target freezing [81]. Furthermore, cryoablation is suitable for treating small tumors but has little effect on large ones [82]. Hence, strategies to overcome these limitations are under active development to improve the clinical benefits of this convenient and cost-effective method. Increasing evidence has indicated that, by activating anti-tumor immune responses, cryoablation can work synergistically with drug-loaded nanoparticles to achieve desired therapeutic effects. Wang and colleagues developed a cold-responsive HCPN-CG nanoparticle composed of hyaluronic acid (HA), chitosan, PNIPAM-B and pluronic F127 (PF127) for cold-controlled co-delivery of chemotherapy (CPT) and a photothermal agent (ICG) to human breast tumors in situ [83]. When being cooled below 10 °C, this nanoparticle disassembles and releases the encapsulated agents to kill the tumor cells in co-ordination with cryoablation. Yao and colleagues prepared a simple and biocompatible chitosan-tripolyphosphate (CS-TPP) nanoparticle carrying trehalose, a cryoprotectant [84]. Trehalose can protect natural killer (NK) cells near the target tumor from cryoinjury, preserving and enhancing the anti-tumor immunity following cryoablation (Figure 5). Notably, this method is capable of preserving surrounding healthy tissues as well. Chen and colleagues utilized PLGA to co-encapsulate an ICG and R837, a Toll-like receptor (TLR)-7 agonist, and an immune checkpoint inhibitor [59]. This nanoparticle can release R837 when stimulated by photothermal therapy, enhancing TAA-elicited anti-tumor immunity in mice. Notably, nanoparticles alone can change the mobility and penetrability of the tumor cell membrane, sensitizing the cells to cryoablation [85].

### 2.6. Photothermal Ablation

Photothermal ablation (PTA) employs photothermal conversion agents to convert benign electromagnetic radiation, preferably near infrared (NIR) light, into heat to ablate cancer cells. The challenges PTA encounters include the limited penetration depth of the commonly used NIR-I laser (750–1000 nm), which makes PTA ineffective for deep-seated tumors, and the limited efficacy of PTA as a standalone therapy. Similar to other ablation therapy, PTA triggers the in situ release of TAAs and damage-associated molecular patterns (DAMPs) to elicit immune responses [86]. However, the immunosuppressive TME counteracts PTA-induced anti-tumor immunity. To overcome this problem, several therapeutic methods are used to boost PTA [87]. In this review, we focus on nanoparticles as drug carriers or immune adjuvants that work in synergy with PTA to achieve satisfactory therapeutic results. Yu and colleagues constructed polyethylene glycol-stabilized platinum nanoparticles conjugated with a PD-L1 inhibitor (BMS-1) via a thermo-sensitive linker. Upon exposure to NIR radiation, BMS-1 was released, and the nanoparticles captured TAAs from the ablated tumor cells, enhancing their internalization and presentation [56]. The nanoparticles also acted as immune adjuvants to enhance DC maturation. In addition, BMS-1 attenuated T cell depletion and promoted the infiltration of effector T cells into tumor tissues (Figure 6A,B). The microenvironment of solid tumors is usually characterized by hypoxia and weak acidity from lactic acid accumulation caused by hypoxia-driven metabolic reprogramming. Since lactic acid in the TME promotes tumor cell immune evasion by suppressing immune cell activity, increasing the metabolism of lactic acid is considered a plausible strategy for improving oncotherapy outcome. To this end, Zheng and colleagues constructed LOX-loaded, Cu^2+^-chelated, PEG-modified mesoporous polydopamine (PDA) nanoparticles (mCULPs). Upon PTA exposure, mCULPs release LOX, which consumes lactate to H_2_O_2_. The resulting H_2_O_2_ is catalyzed by Cu^2+^-chelated mesoporous PDA to produce oxygen, alleviating hypoxia in the TME and enhancing anti-tumor immunity [57]. Liang’s study proposes a nanocomplex that delivers CDDP and PD-L1-targeted DNA aptamers to NSCLC tumors, releasing therapeutic cargos and Fe^2+^ under the acidic tumor microenvironment. The nanocomplexes generate hydroxyl radicals to enhance chemotherapeutic efficacy and reverse immunologically “cold” tumors to “hot” tumors, providing a potential treatment paradigm for NSCLC [88]. Li and colleagues synthesized a PDA-based core–shell nanoplatform loaded with CpG ODN immunostimulants to trigger powerful PTA and anti-tumor immune response [58]. The cationized PDA core coated with a hyaluronic acid shell acted as an effective photothermal agent that increased tumor surface temperature by 16 °C and induced ICD (Figure 6C). Chen and colleagues synthesized PLGA-based nanoparticles composed of R837 and the photothermal agent ICG [59]. In NIR-induced laser ablation of primary tumors, the PLGA-ICG-R837 nanoparticles induced a strong immune response (Figure 6E–G). The PLGA-ICG-R837-enhanced PTA in combination with anti-CTLA4 checkpoint blockade immunotherapy suppressed regulatory T cells (Tregs) and inhibited metastasis and tumor recurrence in mice (Figure 6D).

### 2.7. Others

In recent years, several new ablation techniques have emerged in oncotherapy. These new methods, which include NanoKnife, terahertz (THz), electrodynamic, and NIR ablation, can also be boosted by nanomaterials carrying chemotherapy or immunomodulatory agents. The NanoKnife procedure, which is based on irreversible electroporation (IRE), destroys tumor cells by releasing short pulses of high electric field. These pulses of high electric field create numerous irreversible nanoscale perforations on the tumor cell membrane, allowing molecules of different sizes to enter and exit cells freely, causing rapid apoptosis [89]. As NanoKnife is predominantly a nonthermal ablation technique, it can avoid the heat sink effect and prevent thermal injuries of non-targeted tissues such as blood vessels [90,91]. Recent studies have shown that IRE leads to the in situ release of large amounts of TAAs, which can act as tumor vaccines to induce anti-tumor immune response after ablation to kill the remaining tumor cells and inhibit local recurrence and distant metastasis. When used in combination with immunotherapy, NanoKnife can achieve synergistic effects against malignant tumors, with a broad application prospect in oncotherapy [92]. THz radiation (also referred to as THz waves) with frequencies ranging from 0.1 to 10 THz is widely used in military and security, as well as in medical imaging, diagnosis, and treatment. As an emerging new tool in medical diagnosis and treatment, THz radiation has considerable advantages over other more conventional radiation techniques. Firstly, unlike X-rays, THz radiation does not ionize biomolecules, so it does not damage human keratinocytes [93]. Secondly, the vibration, rotation, and oscillation of biomolecules and hydrogen bonds (which are abundant in water-based biological samples) are well spaced at THz frequency [94]. Thirdly, due to its much longer wavelength than that of visible light or infrared radiation, THz radiation propagates through biological tissues with much less scattering loss. The molecular spectra of molecules involved in cell metabolism (e.g., NO, CO, O_2_, CO_2_, and •OH) are within the THz frequency range [95]. THz radiation is strongly absorbed by polar molecules such as water. The higher water content of cancer cells than normal cells make THz radiation a suitable method not only for detecting cancer cells, but also for destroying them. As such, THz radiation has been used to shrink tumors, and strong THz radiation has been used to treat skin cancer [96,97]. In electrochemical therapy, multiple electrodes are directly inserted into the tumor, and a direct current is applied to drastically change the pH value near the inserted electrodes, killing the surrounding cancer cells. The limited effective area of electrochemical therapy hinders its clinical application. To overcome this problem, platinum nanoparticles are introduced to catalyze the production of toxic reactive oxygen species (ROS) under oscillating electric field, in a procedure called electrodynamic therapy. Electrodynamic therapy exhibited an enhanced anti-tumor efficacy in a variety of cancers [98,99,100,101]. As a standalone therapy, electrodynamic therapy can effectively ablate tumors in ten minutes. However, achieving long-term tumor inhibition remains a great challenge, and single-session therapy can even increase the risk of tumor recurrence after treatment [102,103,104]. The combination of electrodynamic therapy and immunotherapy can reduce the recurrence rate to some extent. For instance, Chen and colleagues incorporated a chloride ion transporter (CIT) into fine Pt/Cu alloy nanoparticles (PtCu_3_ NPs) for enhanced electrodynamic therapy [105]. CIT can increase intracellular Cl^−^ concentration by transporting extracellular Cl^−^. With the help of CIT, these modified nanoparticles can transform endogenous H_2_O_2_ into highly cytotoxic •OH, deplete glutathione, and produce ROS effectively under an electric field. Electrodynamic therapy in combination with these nanoparticles achieved excellent anti-tumor efficacy in vitro and in vivo for relatively large tumors with a volume of approximately 500 mm^3^. Chen’s research team also incorporated 6-diazo-5-oxo-l-norleucine (DON), a glutamine antagonist, into Pt-Pd nanoparticles [60]. DON can prevent the production of glutathione, leading to decreased ROS clearance and increased ROS accumulation during electrodynamic therapy. DON synergized with electrodynamic therapy to induce DC maturation, CD8^+^ T cell infiltration, and ICD, as well as the combined therapy demonstrated great efficacy inhibiting tumor growth, metastasis, and recurrence in animal models. Finally, NIR light can also be combined with nanomaterials for tumor ablation. Hirsch and colleagues first reported thermal therapy using a class of nanoparticles called metal nano-shells [106]. The metal nano-shells are composed of a spherical dielectric core of silica and a thin shell of gold and have tunable optical resonances [107,108]. By tuning the nano-shells to strongly absorb NIR light, the nano-shells can be used to transform extracorporeally applied NIR light into heat energy for tumor ablation. In in vivo studies, exposure to NIR light in subcutaneous tumors injected with metal nano-shells raised tumor temperature by 37.4 °C and caused irreversible thermal damage in tumor cells [106].

## 3. Conclusions and Prospects

Compared with conventional surgical approaches in oncotherapy, tumor ablation techniques have the advantages of reduced invasiveness and pain, reduced adverse effects, faster recovery, simpler procedure, and greater cost-efficiency. However, ablation as a standalone therapy often suffers insufficient efficacy for larger tumors and high recurrence rate. Nanoplatform-based immunotherapy can work synergistically with ablation and amplify ablation-induced anti-tumor immunity. When used together in clinical practice, the appropriate combination therapy has the potential to eradicate primary tumors and prevent tumor metastasis and recurrence.

## Data Availability

Not applicable.

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
