# Peer review of "Nanoplatforms Potentiated Ablation-Immune Synergistic Therapy through Improving Local Control and Suppressing Recurrent Metastasis"

_pharmaceutics, 2023, doi:10.3390/pharmaceutics15051456_

Round 1

Reviewer 1 Report

The review of Wei and colleagues summarised the recent data on ablation therapy and the text provides recent evidences of the alternative procedures. However, the text could not be accepted in the present format.

Major points :

i) the authors described the diverse alternative of ablation therapies based on the recent results, but they did not precise the experimental models used (in vitro, in vivo which animal model etc.) This information must be stated and could be easily added in table 1.

ii) The authors did not exclusively describe the diverse ablation therapies. In fact they related the ablation therapy to the anti-tumoral immune response. It is however not clear whether they referred to the ascobal effects or they hypothesized that the response occurs concomitantly with tumour treatment or not. Any supplement information should be added in the introduction section

iii) The figures are not suitable for publication because of low quality and the embedded text could not be read ( difficult to relate figure to the legend text!). In addition, the legends must provide the information needed for a comprehensive reading; this includes the definition of each abbreviation

Author Response

Thank you for your thoughtful comments on our manuscript " Nanoplatforms Potentiated Ablation-Immune Synergistic Therapy through Improving Local Control and Suppressing Recurrent Metastasis". We appreciate your feedback and suggestions for improvement, and we will do our best to address your concerns.
i) the authors described the diverse alternative of ablation therapies based on the recent results, but they did not precise the experimental models used (in vitro, in vivo which animal model etc.) This information must be stated and could be easily added in table 1.
Response: We apologize for not including the used experimental models in our review. We agree that this information is essential and have added to Table 1.
 ii) The authors did not exclusively describe the diverse ablation therapies. In fact they related the ablation therapy to the anti-tumoral immune response. It is however not clear whether they referred to the ascobal effects or they hypothesized that the response occurs concomitantly with tumour treatment or not. Any supplement information should be added in the introduction section  Response: We apologize for any confusion caused by our wording. We have revised the abstract and introduction section to clarify that our review focuses. In this review, we discussed recent advances in nanoplatform-potentiated ablation-immune synergistic tumor therapy, and discussed the advantages and challenges of the corresponding therapies and propose possible directions for future research.
 iii) The figures are not suitable for publication because of low quality and the embedded text could not be read (difficult to relate figure to the legend text!). In addition, the legends must provide the information needed for a comprehensive reading; this includes the definition of each abbreviation

Response: We apologize for the low quality of the figures in our manuscript. We have made the necessary changes to enhance the quality of the figures and have made sure that the embedded text can be read easily. We have also revised the legends to provide a more comprehensive reading, including the definition of each abbreviation.
Once again, thank you for your valuable comments and suggestions. We believe that the revisions we have made will enhance the clarity and quality of our manuscript. We hope that we have addressed satisfactorily all the concerns in our revised version. If you have any further questions, please do not hesitate to contact us.

Reviewer 2 Report

It is a great review about nanoplatforms Potentiated Ablation-Immune Synergistic Ther-2 apy through Improving Local Control and Suppressing Recur-3 rent Metastasis.

I suggested several changes to authors:

1. Please, you can see the instructions for authors due to that references cited in the manuscript should be cited with [ ].

2. Please, you can see the instructions for authors due to that all references in the reference section is wrong cited.

Figures are very good to clarify the review

Congratulations.

Author Response

Thank you for your positive feedback on our manuscript "Nanoplatforms Potentiated Ablation-Immune Synergistic Therapy through Improving Local Control and Suppressing Recurrent Metastasis." We appreciate your helpful comments, and we are glad to hear that you found the figures to be useful in clarifying our review.
1. Please, you can see the instructions for authors due to that references cited in the manuscript should be cited with [ ].
Response: We apologize for the mistakes in the referencing style in our manuscript. We have carefully reviewed the instructions for authors and made the necessary changes to ensure that all references cited in the text are in the correct format, including using square brackets [ ] as instructed.
2. Please, you can see the instructions for authors due to that all references in the reference section is wrong cited  

Response: We have also carefully checked the reference section to ensure that all references are correctly cited. If you could kindly provide more specific details on the errors you have found, we will promptly make the necessary corrections.
Once again, thank you for your feedback and your kind words. We hope that the revisions we have made address your concerns, and we look forward to hearing back from you soon.

Reviewer 3 Report

The review data are valuable

If you can adding more tables comparing different ablations application using different teqhnique

Author Response

The review data are valuable If you can adding more tables comparing different ablations application using different teqhnique

Response: Thank you for your valuable feedback on our manuscript. We are pleased to hear that you found our data valuable. We have carefully considered your suggestion and agree that adding more tables comparing different ablation applications using different techniques would provide valuable information to readers. Therefore, we have added additional tables (Table 2) to the manuscript that compare the different ablation techniques used in various studies. These tables highlight the strengths and limitations of each technique and provide a comprehensive overview of the current state-of-the-art in this field. We believe that these tables will be useful to researchers interested in this area of study and will enhance the overall value of our manuscript.
Thank you once again for your helpful comments, and we hope that our revisions meet your expectations.

Reviewer 4 Report

The Review " Nanoplatforms Potentiated Ablation-Immune Synergistic Therapy through Improving Local Control and Suppressing Recurrent Metastasis" authored by Cuixia Lu et al. reports the report recent advances in multifunctional nanoparticles and nano pesticides. The authors discussed recent advances in nano platform-potentiated ablation-immune synergistic oncotherapy, focusing on common ablation techniques including radiofrequency, microwave, laser, high-intensity focused ultrasound ablation, cryoablation, and magnetic hyperthermia ablation. However, the review is well-organized. The review should be accepted after minor revision.

I have the following points.

(1) The importance of nanoplatforms potentiated ablation-immune synergistic therapy was not well-presented clearly in the introduction part. The authors are requested to add more information and cite relevant references.

(2) Also, check the table and cited references several times.

(3) The authors suggested, “Table 1 could be tabulated with more relative examples which may help the scientific community for extensive research in this area”.

(4) Also, the review has so many mistakes e.g Ca2+ and MnO2, these should be written as Ca2+ and MnO2, respectively. The authors are requested to correct all such kind of mistakes.

(5) I really do not understand the caption of Table 1 “This is a table. Tables should be placed in the main text near to the first time they are cited”.

(6) The authors are requested to make consistency in the review e.g Figure- somewhere it is written as Fig and somewhere it is written as Figure.

(7) Figure 1 could be made more visible because Fig. 1G, H, and I are very blurred.

Author Response

Thank you for your valuable comments on our manuscript "Nanoplatforms Potentiated Ablation-Immune Synergistic Therapy through Improving Local Control and Suppressing Recurrent Metastasis". We appreciate your time and effort in reviewing our work. We have addressed your concerns as follows:  We have revised the introduction section to highlight the importance of nanoplatforms potentiated ablation-immune synergistic therapy and included relevant references to support our argument.

(1) The importance of nanoplatforms potentiated ablation-immune synergistic therapy was not well-presented clearly in the introduction part. The authors are requested to add more information and cite relevant references.
Response: We have revised the abstract and introduction section to emphasize the importance of nanoplatforms potentiated ablation-immune synergistic therapy.
(2) Also, check the table and cited references several times.
Response: We have thoroughly checked the table and cited references to ensure accuracy.
(3) The authors suggested, “Table 1 could be tabulated with more relative examples which may help the scientific community for extensive research in this area”.
Response: We have added more examples to Table 1 to provide more information to the scientific community and help in extensive research in this area.
 (4) Also, the review has so many mistakes e.g Ca2+ and MnO2, these should be written as Ca2+ and MnO2, respectively. The authors are requested to correct all such kind of mistakes.
Response: We have carefully proofread the entire manuscript. We have corrected all the mistakes pointed out by the reviewer, including Ca2+ and MnO2.
(5) I really do not understand the caption of Table 1 “This is a table. Tables should be placed in the main text near to the first time they are cited”.
Response: We apologize for the confusion regarding the caption of Table 1. The tables have been inserted near the first citation.
(6) The authors are requested to make consistency in the review e.g Figure- somewhere it is written as Fig and somewhere it is written as Figure.
Response: We have ensured consistency in the use of the term "Figure" throughout the entire manuscript.
(7) Figure 1 could be made more visible because Fig. 1G, H, and I are very blurred.
Response: We have enhanced the visibility of Figure 1, particularly the blurred sections (G, H, and I).
Once again, we appreciate your comments, which have helped to improve the quality of our manuscript. We hope that the revised version meets your expectations.

Round 2

Reviewer 1 Report

Dear,

Only two minor points:

- defined RF in text line 125  page 6

- in the text ( beginning from page 6), the authors used the abreviation  RF treatment, not RFA treatment. Please check that the abreviation used is correct in the text .

Best regards

Author Response

Thank you for your email and for bringing those two minor points to my attention. I appreciate your efforts in ensuring the accuracy of our manuscript " Nanoplatforms Potentiated Ablation-Immune Synergistic Therapy through Improving Local Control and Suppressing Recurrent Metastasis".

We have reviewed the entire text, and we can confirm that the abbreviation RFA is used consistently throughout the manuscript to represent Radiofrequency Ablation.

I would like to thank you once again for your valuable input, and if you have any further comments or concerns, please do not hesitate to contact me.

 Best regards

Reviewer 2 Report

Thank you for your correction.

It is OK for me

Best regards

Author Response

 Response: Thank you for your recent correspondence. I appreciate your message and am glad to hear that everything is okay.
 If you have any further questions or concerns, please do not hesitate to reach out to me. I am always here to assist you in any way I can.
 Best regards  
